# Telemedicine in the COVID-19 Era: A Narrative Review Based on Current Evidence

**DOI:** 10.3390/ijerph19095101

**Published:** 2022-04-22

**Authors:** Giulio Nittari, Demetris Savva, Daniele Tomassoni, Seyed Khosrow Tayebati, Francesco Amenta

**Affiliations:** 1School of Pharmaceutical Sciences and Health Products, University of Camerino, Via Madonna delle Carceri, 9, 62032 Camerino, MC, Italy; khosrow.tayebati@unicam.it (S.K.T.); francesco.amenta@unicam.it (F.A.); 2Plastic Reconstructive and Aesthetic Surgery, Nicosia General Hospital, Nicosia-Limassol Old Road 215, Nicosia 2029, Cyprus; drsavvademetris@gmail.com; 3School of Biosciences and Veterinary Medicine, University of Camerino, Via Gentile III da Varano, 62032 Camerino, MC, Italy; daniele.tomassoni@unicam.it; 4Research Department, International Radio Medical Centre (C.I.R.M.), Via dell’Architettura 41, 00144 Rome, RM, Italy

**Keywords:** telemedicine, e-health, COVID-19, social distance, remote treatment, public health emergency

## Abstract

During the recent COVID-19 pandemic, healthcare providers have been encouraged to increase their use of telemedicine and to adopt telemedicine platforms for the majority of their clients who have chronic illnesses. Due to the outbreak itself, almost all countries worldwide were placed under emergency lockdowns. In this paper, we reviewed the literature regarding the use of telemedicine during the COVID-19 pandemic. Consequentially, we identified the adoption of telemedicine in various countries worldwide and evaluated their future steps in order to increase the adoption of e-health technologies. As a result of COVID-19, the e-health agenda, especially telemedicine, has been accelerated in several countries. COVID-19 is affecting individuals’ daily lives and has created major difficulties in the management of healthcare facilities for both infected and non-infected patients. A large portion of the rapid increase in the use of telemedicine can be attributed to evidence from previous pandemics as well as progress made by the field in response to COVID-19, especially in industrialized countries. A lack of effective treatment, large numbers of unvaccinated individuals, as well as social distancing and lockdown measures suggest telemedicine is the safest and most appropriate way of working with patients and doctors. In spite of this willingness, a large number of barriers need to be overcome in order for the telemedicine system to function properly and effectively throughout countries. In order for telemedicine to be sustainable and beneficial beyond the pandemic, several technical, educational, infrastructure, legal, and economic issues must be addressed and solved.

## 1. Introduction

Telemedicine is a rapidly evolving medical service nowadays. It is defined as the use of devices along with technological resources to gain access to a patient and their health information to assess, evaluate, and diagnose them, as well as to decide if a visit to a healthcare center is required [1]. Therefore, it is considered a diagnostic method as well as a screening method. Telemedicine originates from ancient Egypt where hieroglyphics and scrolls were used to share information about health-related outbreaks and epidemics [2]. Since then, the inventions of the telegraph, typewriter, telephone, and later, television, have aided in the transmission of information [3]. Today, the advancement of electronic and mobile technologies and the implementation of medical devices making telemedicine very accessible and helpful in many fields of medicine, as summarize in Table 1.

The World Health Organization (WHO) declared the outbreak of SARS-CoV-2 (COVID-19) a public health emergency of international concern on 30 January 2020, and it was deemed a pandemic on 11 March 2020. The outbreak itself caused major lockdowns in almost all countries, which heavily affected them; it is still affecting the daily lives of individuals, and it causes major management problems in healthcare facilities both for infected patients as well as non-COVID-19 routine visits. As a result of evidence from previous pandemics and the advancements made to the field, the use of telemedicine rapidly increased because of COVID-19, especially in big, industrialized countries such as the United Kingdom [4], the USA [5], and China [6]. The lack of an effective treatment and still large proportion of unvaccinated individuals, social distancing, and lockdown measures suggest that telemedicine is the safer way of patient–doctor interaction [7].

## 2. Telemedicine in the COVID-19 Era

The use of telemedicine in the COVID-19 pandemic is aiding in improving the epidemiological control, and therefore management, of clinical cases [8,9]. It is a method of protecting both infected and non-infected individuals as well as physicians [10,11].

Telemedicine is used as a method of triage for COVID-19 or a way of treating all other diseases and chronic conditions without the need to physically attend a healthcare facility [12,13,14].

A triage is made through technological aids for screening suspect COVID-19 cases, dealing with asymptomatic or mildly symptomatic patients that do not need hospitalization, and identifying critically ill patients for further treatment in a hospital setting [15,16]. Furthermore [17,18], telemedicine allows mildly infected physicians to continue practicing from their homes. Most of the telemedicine employed for non-COVID-19 cases mainly involves internal medicine, oncology, geriatrics, cardiology, orthopedics, neurology, surgery, and dermatology, along with their chronic complications [19,20,21].

## 3. Use of Telemedicine in Different Countries

### 3.1. USA

Telemedicine was already in use prior to the COVID-19 era in the USA, but the pandemic resulted in an enormous increase from 13,000/week to 1.7 million/week post-COVID-19 [22]. Koonin et al. described this almost 13,000% increase in telehealth visits in October 2020 in almost all fields and specializations [22]. This increase is mainly due to changes in insurance programs [23]. Medicare, the US’s federal elderly insurance program, established reimbursements and various policies regarding telemedicine and telehealth services [24].

This telemedicine shift has many positive impacts on managing the COVID-19 pandemic, but on the same level, it has highlighted the discrepancies in healthcare regarding race and economic statuses [25]. On the one hand, access to healthcare remotely is advanced through technological resources; therefore, the transmission of COVID-19 risk is decreased, allowing healthcare centers to focus their resources on critically ill patients and meet the increasing demands of the pandemic. However, this access is not widely available to all populations.

A very good example is hypertension monitoring. Prior to COVID-19, primary care for hypertension and follow-up was performed in a hospital setting. Lack of health insurance, lack of access to healthcare facilities, and non-White race were the major risk factors for the poor control of hypertension [26]. Since face-to-face meetings have been stopped due to lockdown measures, remote monitoring of blood pressure was put into action. The discrepancy in smart phones, home monitoring, and internet access worsens the control, and therefore the monitoring, of this chronic condition [27]. Larger centers have employed telemedicine in monitoring chronic conditions such as hypertension and diabetes; however Southern states and rural areas cannot keep up with this digital implementation [28]. According to the US Census Bureau, one in every five Americans lives in a rural area, and furthermore, 26% of these people feel that they do not have access to healthcare [29,30]. This is mainly due to an inability to afford care, a lack of insurance, and distance from a healthcare facility.

Telemedicine can be the potential solution in all the above scenarios. Patient education can provide easy access to telemedicine platforms and improve overall health outcomes in the management of chronic conditions. Some barriers need to be overcome, including regulations with insurance companies regarding telemedicine services, the inability of physicians to provide care across states, and the distribution of remote patient monitoring (RPM) devices in rural areas [11].

Telemedicine in the US was also employed for male patients with hair loss or andrological conditions such as erectile dysfunction, premature ejaculation, etc., through various platforms such as Forhims and Roman. Rabinowitz et al. [31] showed that during the pandemic, there was a 66% increase in video telemedicine regarding male sexual medical conditions. This increase shows that male patients are still concerned about their sexual health despite the pandemic, and that through telemedicine, they are able to effectively and safely obtain the care they require without the need to attend an outpatient clinic.

Telemedicine’s success has been proven through various studies. Carrier et al. showed that text messaging as a method of follow-up after colorectal cancer was as successful as an office visit and can potentially replace the post-operative follow-up office visit [32]. Powell et al. interviewed patients and found that people are satisfied with telemedicine for dealing with primary care needs, and it is preferable over office visits [33]. Lastly, Segura et al. proved that a follow-up after appendectomy through a smart phone for post-surgical complications at wound side had a sensitivity of 100% and a specificity of 91.7% [34]. All these examples show the promising aspects of telemedicine in dealing with primary care as well as post-surgical monitoring.

Another disadvantage of telemedicine in the US is the lack of patients’ awareness regarding its ease of use, access, and cost. According to a Telehealth Satisfaction Study, 74% of people believe that their health system does not offer telemedicine services. What is more, 72% of rural consumers are unaware of such services [35].

### 3.2. Latin America

Latin America’s adoption of telemedicine is extremely low compared to the US. Financial resources, biases, and, in some cases such as Chile, physician resistance are the main barriers for this slow implementation [36].

However, an increase in telemedicine in Latin America due to the pandemic is seen despite the barriers mentioned previously. A recent study showed an increase of 20.5% annually until 2026 in telemedicine market value [37]. In Mexico, MXN 10 million of healthcare costs was saved due to telemedicine in 2020. In Brazil, there has been an increase of 76.8% in telephone consultation demands due to the pandemic. In Peru and Ecuador, teleconsultations increased as patients are reluctant to visit in-person healthcare facilities due to pandemic fears [38,39,40].

### 3.3. Argentina

The pandemic was a shock to Argentina’s healthcare system. Electronic prescriptions were not around before, and the pandemic forced the government to pass laws to validate them. Telemedicine before COVID-19 was mainly used by people around 30 years of age with no major health problems. The pandemic forced an increase in telemedicine use, and now, the average telemedicine patient is 65 years old with pre-existing conditions [41].

However, further legal and regulatory measures need to be implemented for patient insurance, privacy, and professional licensing for a successful telemedicine system in Argentina. In September 2021, the Improving Hypertension Control in CHina and ARGEntina with a mobile APP-based telecare system (CHARGE-app study) was launched. The study provides a good evaluation of telemedicine versus classic office care for hypertension management [42].

### 3.4. North Africa

The effect of COVID-19 on African individuals is very different than that on the rest of the world. In July 2021, only 2% of the African population was fully vaccinated [43]. Despite this, the mortality rate was ninety-two per one million individuals. This can be explained by the higher younger population in this part of Africa, insufficient testing, and possible previous exposure to local coronavirus strains. WHO reported that Africa has 24% of disease burden compared to the rest of the world, but only 3% of health workers are working with less than 1% of the world’s health expenditures in this region [44]. Telemedicine can potentially solve Africa’s main health issues, but unfortunately, many barriers exist in the region.

The main issue is the shortage of doctors on top of the bad infrastructure, poverty in the region, irregular power supply, and limited internet connectivity [45].

The COVID-19 pandemic raised awareness for the need to implement telemedicine in Africa’s healthcare system like the rest of the world, but the limitations mentioned previously hinder its use [46]. Most reports of telemedicine during COVID-19 indexed in PubMed are from Nigeria, and these provide very little details on the actual services and mode of telemedicine in the country. What is more, the modes described in these papers are telephone and WhatsApp (Version 2.2.7.74. Google Commerce Ltd.; 1601 Willow Road Menlo Park, CA, USA) something unsurprising regarding the poor technology infrastructure in this region. It is now clear that improving the underlying technological problems will increase telemedicine usage across the country, despite the shortage of doctors, and consequently, this will improve the healthcare system as a whole.

### 3.5. South Africa

South Africa’s telemedicine use was very limited prior to COVID-19 due to the restrictions imposed by the General Ethical Guidelines for Good Practice of the Health Professions Council of South Africa. Telemedicine was restricted for doctor–doctor communication, and a telemedicine doctor–patient consultation was allowed only if an existing doctor–patient relationship was present [47,48].

The health system in South Africa is composed of the state-funded public system with salaried doctors and the private sector with service-based fees. COVID-19 forced telemedicine to be implemented in the private sector as a method of triage, avoiding unnecessary face-to-face contact, and maintaining an income [49]. This was aided through the partial relaxation of the guidelines for telemedicine due to the pandemic that allowed telemedicine consults without a prior doctor–patient relationship [50,51].

These implementations caused new services to be developed and medical aids to be used for telemedicine, relieving the system from a lot of strain. Triage via telemedicine caused 97% of the patients to be dealt with without face-to-face contact, reducing the healthcare facility’s burden. Nurses were also triaging patients in pharmacies and teleconsulting, if necessary, with appropriate medical specialists [49]. A new public service reduced the mortality rate of COVID-19 high-risk diabetic patients by 20% by identifying them at admission and following them up though telephone calls during the next few days [52].

The main issues, though, to be dealt with are the fees that can vary from specialty to specialty and whether a service is provided and whether a video or standard call has been used. Medical records shared in software databases are still a major concern.

### 3.6. Canada

Canada’s telemedicine services offer virtual care delivery (VCD) and remote patient monitoring (RPM) [53,54]. VCD provides video/telephone calls between providers and patients, appointment scheduling, document filing and sharing, and virtual waiting rooms. RPM is used mainly to monitor high-risk patients that have been discharged with chronic health conditions such as diabetes, hypertension, and heart failure [54,55].

Virtual care visits have increased exponentially due to the pandemic as Canadians have a high interaction with the Internet, smart phones, and smart devices [56]. A data analysis study from Ontario found that virtual care use in the area increased from 1.6% in the second quarter of 2019 to 70.6% in the second quarter of 2020 [57]. In another survey in the region of British Columbia, there was an increase from 10% to 80.7% in telephone visit calls for hypertension [58].

Despite the high vaccination percentages in Canada and the easing of restrictions, virtual care seems likable in the population. The government is envisioning a hybrid, flexible model of healthcare that uses digital technology to make a bridge between clinic visits and continuing virtual aftercare. This is beneficial both for patients and doctors. For patients, this means easier access to healthcare records, greater reliance on self-management, which promotes a better quality of life, and increasing the use of home monitoring devices. For doctors, on the other hand, this implies a greater reliance on digital technology for monitoring, prescribing, and following up their patients [59].

A survey performed in May 2020 showed 30% of the Canadian population is considering virtual care as the ideal method of first contact with healthcare providers and 57% of the population is already using telemedicine with a 91% satisfaction rate [60]. Despite the general success of telemedicine in the country, rural areas’ broadband is 15% of that of urban areas [61]. Moreover, one third of the Indigenous population reported that they rarely or never use the internet [62].

To take it one step further, the Canadian Medical Association established the Virtual Care Task Force to identify and deal with the barriers of telemedicine in the country [54].

### 3.7. China

Wuhan, China, was the starting point of the world pandemic of COVID-19 in December 2019. The huge lockdowns in the country encouraged telemedicine use for the prevention as well as management of COVID-19 cases as well as chronic disease management. Three main platform types were implemented during the epidemic. Virtual internet hospitals operated by the public hospitals, virtual internet hospitals operated by online healthcare enterprises, and local digital government platforms [63]. The various services available included COVID-19 consults, consults for pre-existing chronic conditions, medical imaging consultations, consultations for new derived health problems, psychological consulting, screening for suspected COVID-19 cases, and medical assistance robots.

Online consults increased dramatically during the epidemic in the major hospitals of the country; triage for suspected COVID-19 cases and consultations for mild symptomatic patients relieved a huge burden on healthcare facilities. A study among physicians showed a 94.6% use of telemedicine during the epidemic, with 34.1% of them never having used telemedicine before. Their biggest concern was the inability for a physical examination of the patient [64].

During the epidemic, an intelligent Hypertension Excellence Center System was developed to follow up and monitor patients with hypertension [65]. This allowed patients to enter their values in a database app and doctors to monitor them, provide counselling, and prescribe medication though the system with no need for a physical consultation, which decreased the exposure to, and possible risk of, a coronavirus infection.

Shandong is an economically developed province in East China, neighboring Japan and South Korea. It was for this reason that well-organized management of the epidemic in the area was critical to help prevention in neighboring countries. Shandong Health Committee responded very well in managing the epidemic in the province and established the functional Anti-Epidemic Expert Group [6]. This expert group is responsible for formulating diagnoses, treating protocols, and quarantine protocols. They use telemedicine to connect doctors, patients, and stored information. The platform is updated in real time with the latest COVID-19 information and instructions for personal protection and quarantine measures. There is also a 24 h/7 d online consulting clinic that performs primary screening and triage, decreasing the burden of hospitals. Suspected COVID-19 patients, if opting for hospitalization, were advised to arrive to a designated hospital, and after discharge, they were followed up via telemedicine. Positive COVID-19 patients with mild symptoms were monitored remotely via telemedicine. Overall, telemedicine saved the province time and money, limiting the unnecessary exposure of patients to COVID-19.

### 3.8. Japan

Japan showed a rapid decrease in outpatient clinic visits during the pandemic. Chronic patients suffering from diabetes, cardiovascular diseases, and hypertension were reluctant to visit a hospital setting during the pandemic. A survey performed in Tokyo showed that 33% of chronic patients were reluctant to contact their medical doctor in case of cardiovascular disease aggravation [66]. The COVID-19 pandemic caused the Ministry of Health to consider telemedicine as a first consultation measure as well as for prescription reasons in an attempt to decrease the strain on the healthcare system. In June 2021, Japan declared that pandemic measures for telemedicine would be continued permanently. However, many doctors and hospitals are reluctant to use telemedicine for providing healthcare due to the sophisticated telemedicine system. The major form of telemedicine used in Japan is the telephone due to its ease of use.

In 2011, the Disaster Cardiovascular Prevention (DCAP) Network was developed in the area of Minamisanriku to help control blood pressure in hypertensive patients [67]. It is a web-based information technological blood pressure monitoring system that helps with the management of hypertensive patients. During the pandemic, the DCAP system continued to be used, using a web cloud network to upload data, and during the relocation of patients back to their homes, it provided a very useful monitoring system for controlling and managing high blood pressures in a house setting. It was found that the DCAP monitoring system is able to manage home blood pressure ranges at an optimal level without the need for anti-hypertensive medication modifications, even during seasonal changes.

### 3.9. Germany

As expected, the pandemic caused a huge burden on the healthcare system of Germany. Developmental changes took place to embed telemedicine in the system [68]. In 2019, the Digitale Versorgung Gesetz (DVG) was introduced to help transform the healthcare system to a digital mode, and it also contains digital healthcare applications through health insurance companies [69].

A smart phone application was introduced on 16 June 2020 by the German government to include warning information notes and vaccination statuses and to exchange data with similar applications across the European Union [70].

Regulations were also updated to include video consultations in order to decrease face-to-face consultations. Despite an increase in demand, these consultations represent only 1% of total consultations [71]. The North Rhine–Westphalia state established an intensive care hospital for hospital teleconsultation services at the beginning of the pandemic to help telemonitoring, web storage, and video conferencing [72]. All the above show that the system in Germany is able to withstand a telehealth model and implement it as a permanent aid in their health system.

However, telemedicine in Germany is not considered a sustainable solution for all. In a study by Rodler et al. [73], urology cancer patients were asked their perspectives on adapting telehealth during the pandemic. The results showed that patients accept telemedicine during the current pandemic situation for administration purposes, prescription, follow-up consultations, etc., but prefer a face-to-face meeting with their doctor for critical decision making in their treatment. Therefore, we can safely conclude that doctor–patient live interaction is crucial for cancer patients and needs to be included in future adaptations of a telemedicine model, especially for this group of patients.

### 3.10. Australia

The government of Australia was one of the few countries that responded quickly to the pandemic with massive lockdowns and telemedicine services easily available to decrease the physical contact of patients with each other and healthcare professionals. The first two waves were small, but unfortunately, the third Delta variant entered the country and increased positive case numbers dramatically.

The government introduced a mass population telehealth system, which includes video and telephone calls as the primary consultation method within the healthcare system. Despite the readily available smart devices and Internet, physicians prefer the use of standard telephone calls for consultations [74,75].

In an online survey performed with surgeon physicians, 38% described telehealth’s quality of care as equivalent to that of face-to-face meetings. The only drawbacks were the inability to perform an actual physical examination and the inappropriateness of breaking bad news through a call [76,77]. In an outpatient survey, on the other hand, most patients were satisfied with the quality of care of telemedicine and had a similar concern regarding physical examinations, and the majority showed a desire for hybrid model clinics in the future [77].

### 3.11. Hungary

Hungary registered the highest mortality rates in Europe, especially after the second and third waves, encountering 313.32 deaths per 100,000 individuals. This is mainly due to the delayed imposition of prevention measures in the country. The restrictions placed by the health authorities made it difficult to manage chronic conditions due to a lack of healthcare access and the limited telemedicine services available in Hungary. One of the few telemedicine platforms in the country is the National e-Health Infrastructure (EESZT) [78]. The aim of EESZT is to digitalize all data and provide access from the various healthcare hospitals in the country. It is an operating system that allows healthcare professionals, patients, and pharmacies to have access to medical data from different locations. Based on the system, patients with chronic conditions can access and assess their conditions during the pandemic. However, the main problem arising is their inability to measure their stats, including blood pressure, blood glucose values, and electrocardiograms, and send the data to their doctors. One solution is to manually collect the results and send them by mail to providers and then receive an email reply for modifications in medication if necessary. The other option is smart devices and mobile apps that can electronically measure these parameters. However, due to the lack of an effective telemedicine healthcare system in the country, the whole evaluation process is delayed.

The Hungarian Hypertension Registry has special devices for measuring blood pressure in various outpatient clinics and for family physicians. In this way, office blood pressure values are registered on an online evaluating center via the Omron Medistance System [79].

Regarding diabetic patients, the D-cont e-Diary System (DAB Pumps Hungary KFT, H-8800 Nagykanizsa, Buda Erwo u.s HUNGARY) is in action, which includes self-measuring smart devices and an online e-diary. In this way, a diabetologist can easily access and monitor daily, weekly, and monthly analyses of blood sugar levels and can conclude better plans for their patients.

### 3.12. Italy

Italy suffered tremendously from the COVID-19 pandemic. This was mainly due to a shortage of healthcare personnel, monitoring devices, and intensive care unit beds. In April 2020, the Istituto Superiore di Sanità published recommendations for telemedicine services with indications and solutions for the planning and managing of patients [80]. Practical recommendations for the telemonitoring of chronic patients have been released, but they are not put into full practice. Due to the lockdown, more and more people relied on online telemedicine to help manage their conditions, both COVID- and non-COVID-related chronic diseases [81]. The National Health Authority set up special monitoring units, i.e., Unita Speciali di Continuita Assistenziale, in March 2020. These units are composed of physicians that provide home interventions to patients with COVID-19 that do not require hospitalization. This was the first attempt of the government to remotely manage patients at home and decrease the burden of hospital facilities [82]. Telemedicine is used in almost all fields of medicine. A survey in Pavia’s polyclinic in San Matteo in the rheumatology department proved that telemedicine is a feasible and a reliable approach that can be used beyond the pandemic era [83]. Patients were questioned about the use of telemedicine (phone/video calls) during the pandemic and the possibility of continuing telemedicine services after the pandemic is over. In total, 78% were satisfied with its use, and 61% were willing to continue their routine follow-up with telemedicine after the pandemic. Another study was performed by San Luigi Gonzaga University Hospital Urology Clinic in Turin [84]. Patients with benign urological diseases that had their appointments cancelled due to the pandemic during March to May 2020 were selected for this study. The results showed 81.5% of the patients were more concerned about the risk of COVID-19 infection than their disease progress, and 66% of the patients were willing to cancel their appointments until the pandemic is over. A total of 88% of the patients were fully satisfied with the telemedicine approach, but despite these values, patients are not actually ready for a telemedicine shift. A total of 53% of the patients did not have access to a smart phone or computer, and the most favorable tools were the traditional phone call and email for communication with their doctor. Therefore, we can conclude that telemedicine offers great advantages in urology benign conditions, but technological improvements need to be implemented for an actual transition to a telehealth modality.

However, these recommendations did not manage to confine the pandemic. Patients who needed hospitalization were unable to access a free bed in healthcare facilities due to a shortage of beds; patients that needed medication were reluctant to visit local pharmacies, causing the mortality rate and the total number of positive cases to rapidly increase.

### 3.13. United Kingdom

The pandemic also caused a major blow to the provision of care in the United Kingdom. National Health Service (NHS) estimates show a dramatic increase in waiting lists in many specializations. For example, in cardiothoracic surgery, waiting lists for diagnostic or therapeutic surgeries increased to 52,000 people waiting for more than 18 weeks compared to only 28 just before the pandemic started [85]. This outlines the shortage of staff and the relocation of resources due to the pandemic. In primary care, there was a dramatic shift to remote medicine, preferably telephone consultations [86].

NHS tried to combat hypertension by providing more than 45,000 blood pressure monitors to individuals through the BP@Home Scheme (Microlife Health Management LtD, St John’s Innovation Centre, Cowley Road Cambrige CB40WS, Cambridge, UK) [87]. Early data showed that the amount of home blood pressure monitoring is three times more compared to pre-COVID levels [88]. What is more, the Royal College of Obstetrics and Gynaecology provided instructions for the home monitoring of blood pressure during pregnancy. Providers offer applications to allow telemonitoring for values and a database for keeping information [89,90,91].

### 3.14. Switzerland

Switzerland is the motherland of telemedicine. Several big commercial telemedical companies are based in Switzerland [92]. Telemedicine was already thriving before the pandemic with 2.5 million patient contacts per annum [93]. Health insurances offer the so-called TelMed model, which requires a teleconsultation before attending an actual health provider’s office. This dramatically decreased the walk-ins in emergency departments [94]. Therefore, the pandemic did not increase the use of telemedicine but rather boosted the technological solutions available for patients to remain in contact with their general practitioners [95].

Telerehabilitation showed an increase during the pandemic, and it became a standard procedure in many cardiovascular centers. Bern University’s hospital provides telerehabilitation via devices that measure blood pressure and heart rhythm for the remote monitoring and diagnosis of post-surgical cardiac insufficiencies [96]. In this way, patients can be easily monitored, and physicians can make a distant diagnosis and give treatment remotely without the need for patients to attend a clinic setting.

## 4. Conclusions

The benefits of telemedicine are summarized as cost-effectiveness in the long run, a decrease in healthcare centers’ strain from unnecessary face-to-face consultations, a decrease in waiting lines at outpatient clinics for follow-ups, triage of COVID-19 positive patients, and decreasing unnecessary exposure of patients and personnel to the COVID-19 virus. As described extensively, telemedicine use during the pandemic is beneficial regarding patient diagnosis, treatment, and follow-up, but it also beneficial for providers. This was proven through the willingness of patients and doctors to continue its use after the pandemic is over [97,98]. Despite this willingness, lots of barriers need to be overcome for the telemedicine system to function properly and effectively throughout countries.

Technological, educational, infrastructure, legal, and economic issues need to be addressed and solved in order to have a sustainable and beneficial telemedicine system platform beyond the pandemic. Laws on how telehealth should be offered, credentialing, medical malpractice, fraud, and abuse all need to be regulated for a telemedicine model to work properly. Another major criterion is fixing reimbursement issues for patients with their insurance and doctor fees. Since this is still ambiguous, some patients and health care professionals in many countries are a bit reluctant to use and provide telemedicine services. What is more, in some countries, reimbursement fees for telehealth services are not yet implemented in insurance fees, which is a major drawback for its use for both patients and healthcare professionals.

There is a need for further research to be conducted in order to identify ways of reducing these barriers in order to increase the effectiveness of telemedicine and the overall performance of the healthcare system. Future studies can specify the effects of telehealth solutions in the efficacy of hospital performance and identify the advantages and disadvantages of using telemedicine, specifically for chronic conditions such as diabetes, hypertension, cancer patients’ follow-up, etc. Moreover, research needs to be conducted on how to implement telemedicine globally as a means of primary care to reduce the burden on hospital facilities. Lastly, we need further evaluation studies on the satisfaction of patients and providers using telehealth services in the long run.

## Figures and Tables

**Table 1 ijerph-19-05101-t001:** Summary of medical device/equipment used for telehealth purposes by medical specializations.

Area of Specialization	Medical Device/Equipment	Usage	Telemedicine Usage
Pneumonology	Portable spirometer Pulse oximetry Phone/videocall	Diagnose and detect possible lung abnormalities. Self-management at home. Asthma and COPD management.	Connected through smart devices/applications. Transmit values to a database for access from physician. Video/phone/email communication with physician for evaluation of readings.
Cardiology	Electrocardiogram Phone/videocall	Prevention screening. Management of heart failure patients and post-operative follow-up.	Connected through smart devices/applications: can detect atrial fibrillation and early diagnosis of heart failure. Remote patient monitoring through a database from uploading readings. Video/phone/email communication with physician for further evaluation.
Diabetic Outpatient Clinic	Blood glucose meters Phone/video call	Self-monitoring blood glucose levels. Prevention of and treating diabetes.	Connected through smart devices/applications. Transmit values to a database for access from physician. Video/phone/email communication with physician.
Plastic and Reconstructive Surgery	Photographs/video call	Post-operative follow-up. Management of post-operative wounds. Initial pre-operative consultations.	Video/phone/email communication with physician for first consultations, follow-up, and management of post-operative wounds. Transmit values to a database for archive purposes.
Urology	Urine analysis strips Telephone/video call	Self-diagnosed, effective, fast, and reduction of sample analysis at laboratories. Successful analysis of glucose, bilirubin, proteins, ketones, pH, erythrocytes, leukocytes, etc.	Transmit values to a database for access from physician. Video/phone/email communication with physician for further intervention if necessary.
Infectious Diseases COVID-19	Video call/telephone	Screening/diagnosis/therapy	Triage of patients: screening low-risk (advise for home treatments) vs. high-risk (home therapy or advise for hospital care).
Rheumatology	Telephone/video call/email	Follow-up and initial consultation	e-prescriptions, e-consultation, diagnosis, and home therapy instructions to decrease the hospital burden. Video/phone/email communication with consultant doctor.
Internal Medicine	Video/phone call/email	Diagnosis and therapy	e-consent, e-prescriptions, e-consultation, diagnosis, and home treatments if possible to decrease the hospital burden.
Hypertension Outpatient Clinics	Blood pressure meters	Self-management, follow-up, and therapy	Connected through smart devices/applications. Transmit values to a database for access from physician if necessary. Video/phone/email communication with physician for follow-up/alteration of medication.

## Data Availability

Not applicable.

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
