# Peer review of "Telemedicine in the COVID-19 Era: A Narrative Review Based on Current Evidence"

_ijerph, 2022, doi:10.3390/ijerph19095101_

Round 1

Reviewer 1 Report

This is a well written paper on the application of telemedicine during the COVID-19 pandemic. While it provides insights into the current state of application, it lacks several aspects that are addressed by the introduction of telemedicine

1.) Telemedicine especially in the US is also expanded to consumer platforms that offer services remotely (Forhims, Roman). Please comment on this fact. 

2.) In Germany, the effects and limitations of telemedicine have been observed. In a urology-oncology population, it has been observed that in the long run patients favor telemedicine/digital technologies especially for administration. For critical decision making they prefer in-patient visits over teleconsultation in the long run (e. g. Rodler et. al., European Urology Focus 2021)

3.) Please elaborate on the different application of telemedicine depending on requirements of the healthcare system in more detail. In Germany, the ease of use might lead to application of telemedicine, while in other countries the lack of specialists or distance towards the next physician is crucial.

4.) How does reimbursement play a role in adoption in different parts of the world? Please comment.

5.) Can you comment on what type of studies have to be performed to overcome the current knowledge gap?

Author Response

This is a well written paper on the application of telemedicine during the COVID-19 pandemic. While it provides insights into the current state of application, it lacks several aspects that are addressed by the introduction of telemedicine.

1.) Telemedicine especially in the US is also expanded to consumer platforms that offer services remotely (Forhims, Roman). Please comment on this fact. 

            Addition on male telemedicine platforms is introduced in the manuscript as well as a study regarding      the use of telemedicine for male sexual conditions in the covid pandemic era. 

2.) In Germany, the effects and limitations of telemedicine have been observed. In a urology-oncology population, it has been observed that in the long run patients favor telemedicine/digital technologies especially for administration. For critical decision making they prefer in-patient visits over teleconsultation in the long run (e. g. Rodler et. al., European Urology Focus 2021)

expansion to include uro-oncology patients and their aspect on telemedicine regarding their everyday follow-ups and crucial decision makings.

3.) Please elaborate on the different application of telemedicine depending on requirements of the healthcare system in more detail. In Germany, the ease of use might lead to application of telemedicine, while in other countries the lack of specialists or distance towards the next physician is crucial.

            Telemedicine in north Africa can be the solution of their main health system but as we mentioned           many drawbacks are still blocking a good implementation of the system including internet                    connectivity, power supply and poverty. However, in most countries, mainly developed, it is proved from the high percentage of patients turning towards telemedicine options that telemedicine is a                    favorable and sustainable evolution of health systems.

4.) How does reimbursement play a role in adoption in different parts of the world? Please comment.

             further clarification on the drawbacks of telemedicine including reimbursement issues, regulatory           laws are outlined in the manuscript.

5.) Can you comment on what type of studies have to be performed to overcome the current knowledge gap?

            further studies are specified in the conclusion section highlighted that are able to improve the                 performance of telemedicine and overcome this knowledge gap.

Reviewer 2 Report

This paper is interesting in the point that readers can imagine the status of telemedicine adoption in several countries under COVID-19. 
Description of this paper was proper in easy reading. But since sorts of adopted tools as telemedicine were too various in independent sections, 
it was difficult to capture overall device concepts. I think that in the introduction section, an introduction of telemedicine method and device 
is needed for clear description. 

Specific comments are as follows:

1. When searching the word of 'diagnosis' in your manuscrpt, I can find two words of line 231, 356 in section 3. But in section of conclusion, the word
of 'diagnosis' functions majorly in line 362. I think that there are ambiguous aspects in this word usage of overall manuscript. 

2. In line 349, the phrase of 'didn't increase the actual use of telemedicine' is difficult to understand. 

3. In Italy section, there is no description of telemedicine usage in this country. This manuscript has been introduced as a Italy resource. This makes no sense in my 
opinion. 

4. I think that there are ambiguous descriptions between COVID-19 related devices and other devices 
(as example, for chronic patients suffering from diabetes, cardiovascular diseases and hypertension).  
If there is any table for device description, this will make better reading. 

5. In line 185, authors skipped inserting % symbol after 1.6.

6. (line 71) In the original paper of Brotman et al. I can't find a sentence related with '3000% increase' phase, although it could be 
a problem spending too a short time in reading this paper.   

Author Response

This paper is interesting in the point that readers can imagine the status of telemedicine adoption in several countries under COVID-19. 
Description of this paper was proper in easy reading. But since sorts of adopted tools as telemedicine were too various in independent sections, 
it was difficult to capture overall device concepts. I think that in the introduction section, an introduction of telemedicine method and device 
is needed for clear description. 

A table is added at the end of the manuscript that sumarizes in detail all medical device and equipment used for telemedicine purposes according to specialization.

Specific comments are as follows:

  1. When searching the word of 'diagnosis' in your manuscrpt, I can find two words of line 231, 356 in section But in section of conclusion, the word of 'diagnosis' functions majorly in line 362. I think that there are ambiguous aspects in this word usage of overall manuscript. 

Medical devices are used to diagnose or treat medical conditions. Diagnosis of a disease through the covid era can be done through telephone/videocall with the practitioner with or without the aid of medical devices. Therefore, diagnosis through covid can be perfomed through telemedicine decreasing in this way the burden to hospital fascilities.

Further expansion of the diagnosis entries have been added through the manuscript to explain how a diagnosis can be performed remotely through medican equipement and video/telephone call, so it is clear for the reader.

  1. In line 349, the phrase of 'didn't increase the actual use of telemedicine' is difficult to understand. 

            The section was rephrased to provide a better understanding.

  1. In Italy section, there is no description of telemedicine usage in this country. This manuscript has been introduced as a Italy resource. This makes no sense in my 
    opinion. 

Expansion on Italian models to include specific studies done in various departments including rheumatology and urology, to potray the advantages and disadvantages of telemedicine use.

  1. I think that there are ambiguous descriptions between COVID-19 related devices and other devices 
    (as example, for chronic patients suffering from diabetes, cardiovascular diseases and hypertension).  
    If there is any table for device description, this will make better reading. 

A table has been added for summarizing all measuring devices for telehealth for a better reading. Summarized expalantion according to each device, its usage and its application in the telemedicine field, sorted by specialization.

  1. In line 185, authors skipped inserting % symbol after 1.6.

            % sign added

  1. (line 71) In the original paper of Brotman et al. I can't find a sentence related with '3000% increase' phase, although it could be a problem spending too a short time in reading this paper.   

The % was from the previous reference Koonin et al (1 700 000-13 000/13 000 x 100%) = 12 976% - corrected to 13 000 %

Round 2

Reviewer 2 Report

Authors responded properly to comments which I proposed.

I think that this manuscript is well written.